# LP-DIXIT: Evaluating Explanations of Link Predictions on Knowledge Graphs using Large Language Models

## Abstract

Link prediction methods predict missing facts in incomplete knowledge graphs, often using embeddings to enhance scalability. However, embeddings complicate explainability, which is crucial for users' understanding of inferences in many domains. Methods emerged to explain predictions by identifying supporting portions of knowledge. To evaluate explanations from a user perspective, they can be compared to those in benchmarks, though they are limited to simplistic graphs. In contrast, user studies on forward simulatability variation measure how explanations improve predictability, i.e., the user ability to predict the results of inferences, which is key to trust. However, user studies face scalability and reproducibility issues on large graphs. Recognizing these gaps, we propose LP-DIXIT to algorithmically evaluate explanations of link predictions by determining forward simulatability variation and adopting large language models to mimic users, as is done in other domains, e.g., in evaluating other approaches on language related tasks. We experimentally prove that LP-DIXIT evaluates as effective explanations those in benchmarks, and we adopt it to compare state-of-the-art explanation methods.

## Keywords

Knowledge Graphs, Large Language Models, Link Prediction, Explanation

**ACM Reference Format:**
Anonymous Author(s). . LP-DIXIT: Evaluating Explanations of Link Predictions on Knowledge Graphs using Large Language Models. In . ACM, New York, NY, USA, 11 pages.

## 1 Introduction

*Knowledge Graphs* (KGs) provide an explicit representation of knowledge in an interlinked and structured manner, enabling knowledge to become not only human-readable, but also machine-readable [25]. KGs are essentially multi-relational graphs composed of entities and relations represented as nodes and edges, respectively. They are known for enabling inference capabilities through integration with web ontologies, which formally define various constructs such as classes and relationships between them. As a result, several KGs have been constructed by both industry [16, 47] and open source communities [4, 9, 38].

Despite their proven utility, the inherent incompleteness of KGs due to their typically incremental and distributed development process [25] often makes inference more complex, also as a consequence of the *Open World Assumption* (OWA) made by default in Logic, which is better suited to web-scale scenarios. To address this problem, the two tasks of *Link Prediction* (LP) and *Triple Classification* have gained importance: they aim to complete KGs by inferring missing facts and determining the truth of given assertions, respectively.

In this paper, we focus on LP methods grounded in *Knowledge Graph Embeddings* (KGEs) [42], as these show impressive scalability. KGEs are vectors in low-dimensional spaces obtained by means of *Machine Learning* solutions to represent elements in the KGs: complex tasks can be solved through simple linear algebra operations on the embeddings. However, embeddings are difficult to relate back to the semantics of the original KG, making KGE models "opaque boxes" whose predictions are difficult to explain, undermining their trustworthiness. This problem severely hampers the application of KGEs, especially in areas where LP is involved in critical decisions, such as finance, healthcare, or autonomous driving. For example, the detection of traffic participants (e.g., vehicles, pedestrians, etc.) can be framed as an LP problem on a KG representing driving scenes [53]. In the case of accidents involving autonomous vehicles, explanations can significantly help to understand their causes in order to prevent them and to manage legal and ethical responsibilities.

Several solutions have been proposed to eXplain LP on KGs (LP-X) [2, 43, 60] in the field of *Explainable Artificial Intelligence* (XAI) [20], which aims to improve the transparency and comprehensibility of ML models, thus making them also more trustworthy. In XAI, methods can be divided into two categories: a) *post-hoc* methods, that compute explanations after the predictions, and 2) *clear box* methods, that produce predictions along with their explanations [20]. We focus on methods in the former category because, unlike those in the latter category, they can be applied to any LP approach. Regretfully, quantitatively assessing the validity of these methods and conducting comparative studies is challenging: there is not yet consensus on a standardized evaluation protocol for assessing the quality of explanations, although the need is widely recognized. Indeed, a research agenda for hybrid (human/machine) intelligence has stated that convergence on such a protocol is crucial [1].

This challenge also stems from the complexity of explanation quality, which encompasses multiple dimensions: *content*, *presentation*, and *user* [35]. In terms of *content*, valuable explanations are those that correctly and completely represent the ML model's behavior, that are similar (dissimilar) along similar (dissimilar) inputs and feature simple interactions among their components. For *presentation*, valuable explanations have a format that enhance clarity, e.g., through abstractions, and can be interpreted without excessive effort. Finally, regarding the *user* dimension, valuable

explanations should allow users to build, revise and express mental models of their understanding of the ML model in the context of their goals, and knowledge, thus allowing them to determine whether the model is trustable and useful [24]. The *user* dimension is crucial because even if the explanations satisfy all the desiderata in the other dimensions, it is ultimately the users who determine whether the XAI approaches are truly applicable in practice.

Moreover, this difficulty is also a consequence of the high diversity of existing explanation structures. Indeed, the *post-hoc* LP-X methods compute not only *prototype* explanations, i.e., those consisting of sets of facts, but also those containing ontological axioms and/or logical rules. Nevertheless, LP-X methods are mainly evaluated through *re-training* of the KGEs, i.e., by measuring the influence of the explanations on solving the very same LP task, thus solely considering the *content* dimension and only by accommodating *prototypes*. In contrast, we aim for an evaluation that covers the *user* dimension, and that is able to accommodate different explanation structures. Alternatively, benchmarks [21, 33] that provide ground truth explanations for each prediction can be adopted. Such explanations are generated through constraints/rules hand-crafted to model domain knowledge and are also validated by users. However, currently available benchmarks employ synthetic data or very limited portions of large KGs, and solely encompass *prototypes*. In contrast, we aim for a solution that can be applied to any real world KG.

In this respect, user studies on the *Forward Simulatability Variation* (FSV) (often referred to as FS) of ML models (for tasks other than LP) [22, 34] also cover the *user* dimension and are flexible wrt the structure of explanation. The FSV measures the variation between the predictability (or simulatability) of the ML inferences before and after the provision of explanations [22]. Note that an inference is *predictable* if a (often human) verifier can hypothesize as accurately as possible its output given the same input and without necessarily replicating the same process. Improving the predictability of ML models is crucial, as it reflects how accurately users can form mental models to represent the ML models, and as such it helps to reinforce user trust in the ML model [24].

However, among the various domains where KGs have been adopted several ones are very complex and specific; as a result, user studies on such KGs require highly specialized users. This difficulty extends to reproducibility challenges: it is difficult to ensure that the users, that were difficult to recruit in the first place, are available for follow-up studies that may happen after a significant amount of time. Therefore, we aim to tailor the formalization of the FSV in LP and to make it algorithmic in order to overcome the shortcomings of user studies.

To further our goals, we propose LP-DIXIT. It algorithmically determines the FSV for *post-hoc* explanations of LP by leveraging *Large Language Models* (LLMs), which are computational models for natural language understanding and generation with very large parameter sizes [11]. LP-DIXIT obviates the need for user recruiting as it employs LLMs as verifiers, thus regarding LLMs as proxies that mimic actual users. Moreover, LP-DIXIT does not require hand-crafted rules, as in the creation of benchmarks, and as such it is not limited to simplified data. The potential ability of LLMs to play such a role is gaining recognition in various domains, including the evaluation of (other) LLMs [61] where the LLM employed as evaluator

is asked to grade an LLM response, possibly with respect to a reference solution or to indicate its preference between two answers; the alignment of human evaluations and LLM evaluations is measured on a benchmark and on a crowdsourced platform. In addition, LLMs have been employed to mimic humans in building training data for natural language processing tasks [19], in performing a variety of social science tasks [3], in generating replies to questionnaires about their experience in video games, and more. Notably, in [3] are conditioned to mimic responses of humans with different cultural background and/or personality. Even if LLMs cannot fully replace real users [52], their adoption in this context represents a scalable and reproducible solution for assessments prior to very expensive user studies. Moreover, we consider LLMs appropriate because they accept flexible prompts, allowing inference to be performed on an input consisting of an LP query enriched with an explanation, which is a necessary step in determining FSV. Furthermore, the flexibility of prompts enables LP-DIXIT to evaluate any type of explanation that can be framed in a prompt.

To further motivate the suitability of LLMs as verifiers, we conduct experiments to compare the evaluations of LP-DIXIT against user rated explanations in benchmarks to measure how well LP-DIXIT aligns with human judgments. We also compare different LLM of different parameter size to assess which one is most aligned with human judgment. Furthermore, we address the lack of a comparative study for *post-hoc* LP-X methods through LP-DIXIT. We summarize our main contributions as follows:

- we formalize LP-DIXIT, to the best of our knowledge, the first solution for computing the *FSV* for post-hoc explanations of LP by leveraging LLMs for automating such assessment
- LP-DIXIT covers the *user* dimension of explanation quality and is a flexible framework that can be applied to any KG and that can accommodating explanations with diverse structures
- We measure the alignment of LP-DIXIT with human judgment through existing benchmarks
- we experimentally compare, on several well known KGs, state-of-the-art (SOTA) *post-hoc* LP-X methods through LP-DIXIT.

The rest of the paper is organized as follows. § 2 reviews existing LP-X methods and the most recent and effective approaches to evaluation. § 3 introduces basic notions essential for understanding the paper. § 4 details our proposal, LP-DIXIT. § 5 illustrates the experimental study. § 6 provides an overview of the achievements and challenges and proposes directions for future research.

## 2 Related Works

In § 2.1 we survey the *post-hoc* LP-X approaches along with the methods for evaluating their explanations; in § 2.2 we review evaluation approaches (including tasks beyond LP) focusing on the *user* dimension.

### 2.1 Post-hoc Explanations of Link Predictions

The methods that first emerged [39, 57] explain a prediction by identifying a single fact: a statement: subject, predicate, object.

Specifically, the approach proposed in [57] is based on perturbations, whilst Criage [39] grounds on approximated *Influence Functions* [44], but is constrained to a limited set of facts. In contrast, current methods return *prototypes*. Kelpie [43] employs a novel *post-training* process that supports any KGE model. Kelpie++ [6] ground on *Graph Summarization* to enhance Kelpie's efficiency as well as effectiveness, and presentation of explanations. Similarly, KGEx [5] distills surrogate models on sampled sub-graphs, whilst KE-X [60] identify the facts maximizing the *Information Gain* wrt the prediction; in contrast, KGExplainer [32] adopts greedy search based on perturbations. Notably, GEnI [2] returns explanations consisting of schema axioms and/or facts based on numerical criteria on predicate embeddings, distance functions, or *Influence Functions*; its explanations are also converted into natural language through hand-crafted templates. However, it is restricted to translational and bilinear KGE models. Differently, the method introduced in [7] provides explanation by performing *abduction* based on learned rules.

The *prototypes* are mainly evaluated by *re-training* the KGE model, i.e., by comparing the LP performance of the original model to one trained on a modified KG, where the facts in the explanations have been added/removed (as in [57], Kelpie, Kelpie++, GEnI, KE-X, and [7]), or isolated (as in KGEx, and KGExplainer). This approach supports any KGE model, but it solely covers the *content* dimension and is tailored to *prototypes*, whereas we strive to cover the *user* dimension whilst supporting different explanation structures. In this respect, the explanations by GEnI also undergo manual inspection, which covers the *user* dimension, but requires human effort whereas we aim at an algorithmic evaluation.

Other proposals, such as FR200K [21], FRUNI and FTREE [33], offer benchmarks of explanations generated through hand-crafted constraints/rules. However, in [33] explanations are not evaluated by users, as such this evaluation does not cover the *user* dimension. In contrast, in [21] users rate each explanation in the proposed benchmark based on how *intuitive* it is, which, however, is a subjective notion. Conversely, we frame the evaluation in the FSV framework. Moreover, such benchmarks employ simplistic data and are tailored to *prototypes*, whereas our target is a solution that can be applied to real world KGs.

A different explanation consists of a path from the subject to the object of the prediction. LinkLogic [30] adopts a perturbation based approach, conversely, CrossE [59] and ApproxSemanticCrossE [12] ground on (semantic) similarity of entities and/or relationships wrt the prediction. The similarity-based explanations are evaluated in terms of the number of similar paths connecting similar entities.

Other methods return an explanation different from a set of facts or a path. In [28] logical rules are mined to explain a set of predictions, whilst FeaBi [26] extracts interpretable vectors from embeddings via *Feature Selection*. In [28] the mined rules are evaluated in terms of their performance in *Triple Classification* on the explained predictions and synthetic negatives (false assertions). In contrast, the vectors resulting from FeaBi are compared to vectors learned with an interpretable approach, moreover, the classification performance resulting from training on interpretable vectors or KGEs is compared.

We also mention [8], XTransE [58], and GNNExplainer [56] tailored for *Graph Neural Networks*, although they are *clear box* methods whereas we target *post-hoc* ones. The explanations generated by XTransE are evaluated as in [28], whilst those resulting from GNNExplainer are compared to a ground truth. In contrast, in [8] solely a qualitative evaluation is presented. Finally, SHAP is notable for its task independence, but it often faces scalability issues. Its explanations are often evaluated by performing *Model Parameters Randomization Check*, i.e., by randomizing model parameters and checking whether the explanations change.

## 2.2 Evaluation considering Users

Although the main goal of this paper is an algorithmic approach, we also review key categories of user studies as they are frequent in the *user* dimension. In the firstly emerged studies, participants are asked to rate explanations based on different qualities, e.g., satisfaction, usefulness, intuitiveness, etc., or to choose the best explanation between two alternatives. However, such approaches involve subjective notions. More recent studies [22, 34] overcome this problem by measuring the FSV that is also our target; it frames the evaluation task as a repeated check of predictability; the user solely evaluates predictability, which in turn is evaluated without referring to subjective qualities [35]. In other approaches [18, 36, 49], participants are asked to identify an irrelevant insight introduced into an explanation, valuable explanations should be easy to distinguish from additional noise. Similarly, in [40, 41, 46, 50] participants are asked to identify, with the help of explanations, a property of the model that has been changed, whilst in [41, 48] they inspect explanations to identify detrimental training data.

Recently, proposals for making such evaluation algorithmic to tackle the challenges of user studies. Firstly, in [37, 40] the effort that a user would need to follow the recommendations in the explanation is measured, whilst in other proposals the agreement between diverse XAI methods is measured. In contrast, we aim at measuring predictability improvement, as it is crucial in trust. Alternatively, benchmarks of ground-truth explanations that are generated according to domain knowledge and evaluated by users have been proposed. However, such benchmarks are often limited to simplistic data. Finally, in [23] (a refined version of) the FSV is determined using an LLM to mimic users, but it targets *Natural Language Processing* (NLP) tasks, whereas we focus on LP-X methods. However, in such work the LLM is fine-tuned on examples and explanations of the target NLP task, thus shifting the focus to the LLM's ability to learn the simulation task rather than evaluating the explanations themselves.

## 3 Fundamentals

In this section, we introduce KGs more formally and the basics of KGE methods. A KG is a graph-based data structure $\mathcal{G}(\mathcal{V}, \mathcal{R})$, where $\mathcal{V}$ is a set of nodes representing entities, and $\mathcal{R}$ a set of predicates, representing binary relations between entities. In the adopted RDF model, a KG is a collection of triples of the format $\langle s, p, o \rangle$, statements with a *subject*, a *predicate* and an *object*, where $s, o \in \mathcal{V}$ and $p \in \mathcal{R}$.

Various models have been proposed for representing KGs in low-dimensional vector spaces, by learning for each entity and

predicate in the KG a unique numerical vector (or *embedding*) in a given space. Different types of embedding spaces can be used, such as real, pointwise, complex, discrete, Gaussian, manifold; without loss of generality, we will consider real embeddings in the following, denoting their vectors in bold-face.

These models typically represent each entity $e \in \mathcal{V}$, and each predicate $p \in \mathcal{R}$ by means of an embedding vector $\mathbf{e} \in \mathbb{R}^k$, and $\mathbf{p} \in \mathbb{R}^i$, respectively, where $k, i \in \mathbb{N}$ are hyperparameters. In addition, each model is associated with a *scoring function* $f : \mathbb{R}^k \times \mathbb{R}^i \times \mathbb{R}^k \to \mathbb{R}$: for each triple $\langle s, p, o \rangle$, the score $f(\langle s, p, o \rangle)$ measures the probability of such a statement. In the following, we consider formulations where higher values convey more plausibility; symmetric formulations can be derived for models where lower scores convey higher probability. The embeddings and parameters are learned from the KG by minimizing a loss function based on $f$. To this purpose, the set of triples encoded by $\mathcal{G}$ is divided into a training set $\mathcal{G}_{train}$, a validation set $\mathcal{G}_{val}$ and a test set $\mathcal{G}_{test}$. Besides of entity and predicate embeddings, models can also learn shared parameters that are not explicitly connected to any KG element, similarly to the weights of neural layers.

Given a query for LP, as an incomplete triple $\langle s, p, ? \rangle$, LP is performed by computing

$$o = \arg \max_{e \in \mathcal{V}} f(\langle s, p, e \rangle).$$

In the case of multiple triples having the same scores several strategies can be adopted, e.g., lexicographic tie breaking. Moreover, the *rank* of a triple $\langle s, p, o \rangle$ in $\mathcal{G}_{test}$ is required for evaluating the LP performance and may be defined as:

$$\text{rank}(\langle s, p, o \rangle) = |\{e \in \mathcal{V} \mid f(\langle s, p, e \rangle) >= f(\langle s, p, o \rangle)\}|.$$

## 4 The Proposed Approach

We introduce LP-DIXIT, a method for algorithmically evaluating explanations of LPs on KGs. Specifically, it determines the FSV, which is the variation in the predictability of LP inferences before and after explanations are provided. Note that an LP inference is predictable if (usually human) verifiers can simulate it, i.e., they can provide the same result. To address shortcomings of user studies, LP-DIXIT employs LLMs to mimic human verifiers. In § 4.1 we formally define the FSV in an LP scenario. Next, in § 4.2 we specifically delve into the usage of LLMs as verifiers.

### 4.1 Forward Simulatability in Link Prediction

Given a query $q = \langle s, p, ? \rangle$ for which an LP method predicted the entity $\hat{o}$ as the filler and an explanation $X$ for this prediction, $F_{\hat{o}}$ is a function returning a label $y \in \{-1, 0, 1\}$ indicating that $X$ is, respectively, *harmful*, *neutral*, or *beneficial* in simulating the inference leading to $\hat{o}$[1].

Such a function relies on verifiers whose role is to simulate inferences, ideally they are users since the main goal of XAI is to improve their understanding of ML models, but more generally it can be any other agent capable of performing the simulations, such as an LLM as in our proposed approach LP-DIXIT. In this formalization, we do not specify the nature of the verifier and how

---

[1]We denote $\hat{o}$ as a subscript rather than an argument of the function $F$ because the verifier does not know it

it performs the required computations, but rather denote it as a function $S_{\hat{o}}$ that given a query, or a query with an explanation, for which the LP method predicted $\hat{o}$ as the filler, ideally returns an entity in the KG that the verifier estimates to be the same entity $\hat{o}$ that was predicted by the LP method. We then define $F_{\hat{o}}$ based on such verifier.

Specifically, $S$ returns an entity $\tilde{o} \in \mathcal{V}$ as the simulation of the inference that leads to $\hat{o}$ for the query $q$, formally:

$$\tilde{o} = S_{\hat{o}}(q) \tag{1}$$

To compute this function, the verifier tries to answer the query by relying solely on its prior knowledge, hence we refer to this step as pre-explanation simulation. Similarly, $S_{\hat{o}}$ returns an entity $\tilde{o}_X \in \mathcal{V}$ as the simulation of the inference yielding $\hat{o}$ for the query $q$ given also the explanation $X$, formally:

$$\tilde{o}_X = S_{\hat{o}}(q, X) \tag{2}$$

In this case the verifiers can rely on the explanation in addition to their knowledge, so this step is dubbed post-explanation simulation. Note that, both $\tilde{o}$ and $\tilde{o}_X$ can be equal to or different from $\hat{o}$.

A simulation is correct if the returned entity is equal to $\hat{o}$, hence, the indicator $s_{\hat{o}} \in \{0, 1\}$ denotes if the simulation $\tilde{o}$ is correct or not and is defined as:

$$s_{\hat{o}} = \mathbb{1}_{\hat{o}}(\tilde{o}) \tag{3}$$

Likewise, $s_{\hat{o}}^X \in \{0, 1\}$ denotes the correctness of the simulation $\tilde{o}_X$ and is defined as:

$$s_{\hat{o}}^X = \mathbb{1}_{\hat{o}}(\tilde{o}_X) \tag{4}$$

The correctness of the simulations is based on the filler $\hat{o}$ predicted by the LP method rather than the true filler $o$ (if available) because the FSV involves evaluating the predictability of the model rather than its predictive accuracy. Specifically, evaluating the FSV involves evaluating the ability of the verifier to obtain the same predicted filler $\hat{o}$ rather than the true filler $o$. It is worthwhile to note that for top-ranked triples (correct predictions) $o = \hat{o}$.

Ultimately, $F_{\hat{o}}(q, X)$ returns $y$ as the difference between $s_{\hat{o}}^X$ and $s_{\hat{o}}$, formally:

$$y = F_{\hat{o}}(q, X) = s_{\hat{o}}^X - s_{\hat{o}} = \mathbb{1}_{\hat{o}}(S_{\hat{o}}(q, X)) - \mathbb{1}_{\hat{o}}(S_{\hat{o}}(q)) \tag{5}$$

The values returned by $F$ are to be interpreted as follows:

- $y = 1$: The explanation $X$ lead to a correct simulation when it was previously incorrect (i.e., $s_{\hat{o}} = 0, s_{\hat{o}}^X = 1$), this indicates that the explanation is beneficial for the verifier
- $y = 0$: The simulation correctness does not change, either both are correct (i.e., $s_{\hat{o}} = 0, s_{\hat{o}}^X = 0$) or both are incorrect (i.e., $s_{\hat{o}} = 1, s_{\hat{o}}^X = 1$), this indicates that the explanation was neutral for the verifier
- $y = -1$: The explanation $X$ lead to an incorrect simulation when it was previously correct (i.e., $s_{\hat{o}} = 1, s_{\hat{o}}^X = 0$), this indicates that the explanation was harmful for the verifier

### 4.2 Using a Large Language Model as a Verifier

LP-DIXIT employs an LLM as a verifier to simulate inferences of the LP method, i.e., to compute the pre-explanation simulation ($S_{\hat{o}}(q)$) and the post-explanation simulation $S_{\hat{o}}(q, X)$. For this purpose, it builds a prompt by filling out a prompt template that we engineered and that is shown in Fig. 1. The template features different sections;

An RDF triple is a statement (subject, predicate, object). The subject and the object are entities, and the predicate is a relation between the subject and the object. Perform a Link Prediction (LP) task, specifically, given an incomplete RDF triple (subject, predicate, ?), predict the missing object that completes the triple and makes it a true statement.

Strict requirement: output solely the name of a single object entity, discard any explanation or other text.
Correct format: Italy
Incorrect format: The object entity is Italy.

($\{s\}$, $\{p\}$, ?)

$\{X\}$

**Figure 1: Structured prompt template with sections separated by blank lines and variable parts enclosed in curly braces.**

the first one is a description of the task to be simulated: the first part specifies an RDF triple, then it defines LP as providing the (name of) the entity that fills a query represented as a triple with an unknown object. Such an abstract description enhances the LLM's comprehension of the specific query. The second part of the prompt includes explicit instructions, along with an example, directing the LLM to return only the entity name. Without this specific guidance, the LLM may generate unnecessary additional text, such as introductions, motivations, or invitations to ask for any further clarification. Conversely, the subsequent steps for computing $F$ assume that the simulation will yield only one entity. The last section is the query $q$ to simulate, either with no additional information (when computing Eq. 1) or also relying on the explanation (when computing 2). The prompt's section on the explanation can also contain a hook that describes the general structure of an explanation. LP-DIXIT specifically uses instruction-tuned LLMs, which are obtained by fine-tuning their base versions to datasets containing specific instructions, i.e., detailed descriptions of the input and the task to be performed, because our prompt template contains instructions about the input, the task, and the desired output.

Regrettably, LLMs (as well as other kinds of verifiers) are not aware of the KG on which LP is performed. Specifically, since LLMs are generative models these perform LP by responding with the name of the predicted entity to fill the query in contrast with KGE-based discriminative methods which require to rank all the triples obtained by filling the query with each entity in the KG. Hence, LLMs can generate answers consisting of entities that are not included in the KG. Addressing this gap, LP-DIXIT includes in the prompt a set of entities and a natural language instruction stating that the LLM must pick its response from such set. To clarify, the resulting prompt is akin to those for *Multi-Choice Question Answering* (MCQA) tasks [29], which involve selecting the correct answer from a set of options based on a given question. Ideally, such a set should be $\mathcal{V}$: the set of entities in the KG. Nevertheless, in several

LLMs, the maximum number of tokens (basic units of text processed by LLMs, that can represent words, subwords, or characters) in a prompt is too small to accommodate the entire set $\mathcal{V}$.

To overcome this issue, it exploits the LP method in order to filter $\mathcal{V}$ to a subset $O_q$ that is meaningful for the query $q$, we dub the resulting method variant LP-DIXIT$_O$. Specifically, the LP method fills $q = \langle s, p, ? \rangle$ by first computing the ranked sequence $O$ of all the triples in $\{\langle s, p, e \rangle \mid e \in \mathcal{V}\}$ according to the score function of the underlying KGE model; note that LP-DIXIT$_O$ retains only the object of each ranked triple. Formally:

$$O_q = e_1 \succ \cdots \succ e_n, e_i \in \mathcal{V}, f(\langle s, p, e_1 \rangle) \geq \cdots \geq f(\langle s, p, e_n \rangle) \quad (6)$$

Next, LP-DIXIT$_O$ filters down $O$ to its top $k$ entities to focus on entities relevant to the query, but $k$ should be the largest number of entities that fit in the query to mitigate the bias towards the correct entity. To clarify, we are pursuing a tradeoff: while LP-DIXIT$_O$ focuses on relevant entities, it manages the risk of favoring the correct entity.

One might consider employing *Retrieval-Augmented Generation* (RAG) techniques to ensure that the responses are entities in the KG, as RAG integrates external knowledge sources into the generative process [55]. However, RAG may introduce additional complexity into the model architecture, making it more challenging to maintain and tune and may still face challenges in ensuring that the responses are exclusively entities in the KG. We reserve the exploration of RAG for future studies (as also reported in § 6).

In addition, LLMs may not be able to correctly simulate LPs as they are not trained/fine-tuned for this task or other predictive tasks on KGs. To address this issue, we also developed a few-shot prompt template [31], i.e., a prompt including a set $\mathcal{D}$ of examples (or demonstrations) of solved LP queries, as LLMs are renowned for their in-context learning capability, i.e., learning from examples provided directly in the input without the need for performing expensive fine-tuning. We dub the resulting declination of the method LP-DIXIT$_{\mathcal{D}}$. It selects as examples the triples in $\mathcal{G}$ that are ranked first by the LP method because the correctness of the simulations is determined wrt $\hat{o}$ rather than $o$. It adopts predicate-guided demonstrations, i.e., it selects those queries where the predicate is the same predicate $p$ in order to focus on triples related to the specific query $q$ as in the LLM-based LP method proposed in [54]. Then, LP-DIXIT$_{\mathcal{D}}$ samples $j$ triples from $\mathcal{D}$ and incorporates each triple $\langle s', p, o' \rangle$ in the prompt as the query $\langle s', p, ? \rangle$, along with its corresponding filler $o'$. Finally, LP-DIXIT$_{O\mathcal{D}}$ combines both LP-DIXIT$_{\mathcal{D}}$ and LP-DIXIT$_O$.

## 5 Experimental Evaluation

We illustrate the experimental evaluation that was carried out, specifying the experimental setup and discussing quantitative results.

### 5.1 Experimental Setting

We measured the alignment of LP-DIXIT wrt human judgment in determining the FSV by comparing its output on benchmark explanations to the labels in the benchmark. We evaluated all the variants of LP-DIXIT on the benchmarks FR200K, FTREE, and FRUNI. In FR200K users rated explanations based on intuitiveness; scores are then averaged and normalized in the interval $[0, 1]$ To mitigate the

subjective nature of intuitiveness, we kept only high-scoring explanations because neutral scores can lead to different opinions among users; similarly, low-scoring explanations may be unclear and ambiguous, leading to subjective judgments, whereas high-scoring explanations are likely to be less ambiguous, leading to more consistent judgments. Specifically, we performed quantile based discretization of the scores into the categorical labels $\{-1, 0, 1\}$ and we kept solely the explanations with label 1. In contrast, in FRUNI explanations are not rated by users and we assumed all explanations to have label 0.

Furthermore, we performed the experiments with different LP methods as the FSV can vary depending on the LP method. Specifically, three examples in key families of KGE models were adopted: TRANSE [10] (translational), CONVE [13] (neural) and COMPLEX [51] (bilinear). However, in both benchmarks, explanations are associated to rule-based predictions whilst the prompts including the entity set $O$ and the example set $\mathcal{D}$ rely on KGE-based predictions. To fill this gap, we kept solely the test triples that are ranked first by the LP method and that have a ground truth explanation; we excluded the cases where no top-ranked test triples were available (FTREE with all models and FRUNI with ConvE).

As verifiers, we adopted the SOTA LLMs Llama-3.1 [17] (Llama3.1-8b-Instruct, Llama3.1-70b-Instruct) and Mixtral [27] (Mixtral-7x8B-Instruct-v0.1). We chose $k = 100$ entities in $O$ according to the limit of 8192 tokens in Llama-3.1 and $j = 10$ examples in $\mathcal{D}$ as it is a rather popular choice [31]; we adopted the same values for Mixtral to ensure a fair comparison. We measured the proportion of explanations for which LP-DIXIT correctly returned 1 along with the occurrences of 0 and $-1$, as these represent different types of errors, with 0 being the least severe.

We also adopted LP-DIXIT for comparing SOTA *post-hoc* LP-X methods. Specifically, we adopted the best setting resulting from the evaluation on the benchmark. Such comparative study encompassed several KGs: FB15k-237 [10], WN18RR [10], YAGO3-10 [14], YAGO4-20 [6], DB50K [45], DB100K [15]. FB15k-237 is extracted from Freebase by selecting entities with at least 100 occurrences and removing redundant relationships. Similarly, WN18RR is extracted from WordNet by retaining the entities involved in triples with specific predicates and excluding those appearing in fewer than 15 triples. In addition, DB50K and DB100K are extracted from DBpedia, while YAGO4-20 is extracted from YAGO4 by retaining the triples with entities that appear in 20 or more triples and excluding those with literal objects. YAGO4-20, DB50K, and DB100K, along with RDF triples, incorporate OWL statements (in the OWL2-DL format) including class assertions, and other schema axioms, e.g., disjointness, details of such integration are reported in [6].

The compared *post-hoc* LP-X methods are CRIAGE, the method proposed in [57] (that we dub DP short for Data Poisoning), KELPIE, KELPIE++, and GENI which are recent and effective approaches also supplying data and code. CRIAGE, DP, KELPIE and KELPIE++ can be executed in two different versions: *necessary* (nec) and *sufficient* (suff). For each version KELPIE++ can be run with two different approaches to *Graph Summarization*: bisimulation (b) and simulation (s). However, we executed GENI and CRIAGE exclusively for TRANSE and COMPLEX because they are tailored to translational and bilinear models. Similarly, we executed KELPIE++ exclusively on YAGO4-20,

DB50K, and DB100K because it requires KGs equipped with OWL statements.

Furthermore, for each configuration of model and dataset, we compare the *post-hoc* LP-X methods on a set of 100 randomly sampled test triples that are correctly ranked first by the LP method. It is very important to note that each KGE model will lead to a different set of top-ranked triples. Therefore, the comparison is not intended to be between different KGE models, but rather between different LP-X methods applied to the triples that each model ranks correctly. Each top-ranked triple $\langle s, p, o \rangle$ is regarded as a query $q = \langle s, p, ? \rangle$ along with its filler $\hat{o}$, note that $\hat{o} = o$ for each triple as we focused on correct predictions. Next, we computed the explanation $X$, its label $y = F_o(q, X)$, and its indicators $s_{\hat{o}}, s_{\hat{o}}^X$ for each test triple. We aggregate the results over 100 triples by computing the average $\bar{s}$ of the correctness indicators of the pre-explanation simulation, and the average $\bar{s}^X$ of the correctness indicators of the post-exp simulations; the final score is then computed as $\bar{y} = \bar{s}^X - \bar{s}$ and represents the average FSV. Finally, we adopted the same LLMs and the same values for $j$ and $k$ employed in the evaluation with the benchmark. The code, the datasets, and the trained KGE models utilized in our study are openly accessible on GitHub[2]. In Appendix A, we provide detailed information on the hyperparameters utilized for training the KGE models, as well as those used in the explanations methods, and in the LLMs.

## 5.2 Outcomes of the Evaluation

In Tab. 1 we report the outcomes in terms of percentages of outcomes equal to $-1$, 0, or 1, of the experiments that measure the alignment of LP-DIXIT with human judgment. In such Table L-70B (L-8B) stands for Llama3.1-70B-Instruct (Llama3.1-8B-instruct) and M-7B stands for Mixtral-8x7B-Instruct (in the following we omit Instruct in the name of LLMs).

The percentage of explanations from FR200K that return 0 is consistently higher than the percentage of those that return $-1$. Thus, even if LP-DIXIT does not return 1 for all explanations in FR200K, the majority of errors are of the least serious type. In addition, LP-DIXIT with Llama-3.1-70B as the verifier consistently outperforms the other configurations on FR200K. In contrast, LP-DIXIT using Mixtral-8x7B performs best on FRUNI, as it correctly returns 0 for all the explanations.

It is also worth noting that when using Llama-3.1-70B, which is the largest model in terms of parameter size, the addition of $O$ and $\mathcal{D}$ is almost always detrimental. For instance, on predictions made on FR200K with TransE, when using such LLM, we got the following proportions of 1: 65% for LP-DIXIT, 62% for LP-DIXIT$_O$, 43% for LP-DIXIT$_{\mathcal{D}}$, and 39 % for LP-DIXIT$_{O\mathcal{D}}$. In contrast, in the case of Llama-3.1-8B and Mixtral-8x7B such additions are often beneficial. For instance, on predictions made on FR200K with ComplEx, when using Llama3.1-8B, the sequence of percentages of 1 is 48.2%, 48.9%, 61% , and 65%. It may be that very large LLMs make better use of the input context, such as the explanations we are evaluating. It follows that the use of smaller LLMs is more appropriate when running a very large LLM is too demanding in terms of computational resources. Conversely, if sufficient resources are available, using

---

[2]https://anonymous.4open.science/r/lp-dixit-168D

**Table 1: Alignment of LP-DIXIT with human judgment**

| FSV | LLM | FR200K | | | FRUNI | | |
|---|---|---|---|---|---|---|---|
| | | -1 | 0 | 1 | -1 | 0 | 1 |
| **TransE** | | | | | | | |
| LP-DIXIT | L-70B | 0.025 | 0.311 | **0.664** | 0.000 | 0.834 | 0.166 |
| LP-DIXIT | L-8B | 0.000 | 0.525 | 0.475 | 0.000 | 0.960 | 0.040 |
| LP-DIXIT | M-7B | 0.004 | 0.601 | 0.395 | 0.000 | **1.000** | 0.000 |
| LP-DIXIT$_O$ | L-70B | 0.021 | 0.353 | 0.626 | 0.144 | 0.751 | 0.105 |
| LP-DIXIT$_O$ | L-8B | 0.017 | 0.563 | 0.420 | 0.170 | 0.585 | 0.245 |
| LP-DIXIT$_O$ | M-7B | 0.021 | 0.546 | 0.433 | 0.051 | 0.628 | 0.321 |
| LP-DIXIT$_D$ | L-70B | 0.021 | 0.546 | 0.433 | 0.119 | 0.877 | 0.004 |
| LP-DIXIT$_D$ | L-8B | 0.004 | 0.420 | 0.576 | 0.090 | 0.910 | 0.000 |
| LP-DIXIT$_D$ | M-7B | 0.004 | 0.866 | 0.130 | 0.036 | 0.964 | 0.000 |
| LP-DIXIT$_{OD}$ | L-70B | 0.101 | 0.500 | 0.399 | 0.264 | 0.679 | 0.058 |
| LP-DIXIT$_{OD}$ | L-8B | 0.008 | 0.441 | 0.550 | 0.303 | 0.574 | 0.123 |
| LP-DIXIT$_{OD}$ | M-7B | 0.038 | 0.765 | 0.197 | 0.032 | 0.809 | 0.159 |
| **ComplEx** | | | | | | | |
| LP-DIXIT | L-70B | 0.008 | 0.239 | **0.754** | 0.000 | 0.852 | 0.148 |
| LP-DIXIT | L-8B | 0.002 | 0.516 | 0.482 | 0.000 | 0.961 | 0.039 |
| LP-DIXIT | M-7B | 0.002 | 0.566 | 0.431 | 0.000 | **1.000** | 0.000 |
| LP-DIXIT$_O$ | L-70B | 0.014 | 0.268 | 0.718 | 0.012 | 0.809 | 0.179 |
| LP-DIXIT$_O$ | L-8B | 0.022 | 0.490 | 0.489 | 0.044 | 0.725 | 0.230 |
| LP-DIXIT$_O$ | M-7B | 0.030 | 0.506 | 0.464 | 0.012 | 0.613 | 0.375 |
| LP-DIXIT$_D$ | L-70B | 0.021 | 0.481 | 0.498 | 0.092 | 0.888 | 0.020 |
| LP-DIXIT$_D$ | L-8B | 0.003 | 0.385 | 0.612 | 0.076 | 0.919 | 0.006 |
| LP-DIXIT$_D$ | M-7B | 0.010 | 0.867 | 0.123 | 0.047 | 0.953 | 0.000 |
| LP-DIXIT$_{OD}$ | L-70B | 0.049 | 0.485 | 0.466 | 0.056 | 0.845 | 0.099 |
| LP-DIXIT$_{OD}$ | L-8B | 0.004 | 0.346 | 0.650 | 0.055 | 0.824 | 0.121 |
| LP-DIXIT$_{OD}$ | Mx7B | 0.014 | 0.779 | 0.207 | 0.020 | 0.747 | 0.232 |
| **ConvE** | | | | | | | |
| LP-DIXIT | L-70B | 0.000 | 0.130 | **0.870** | – | – | – |
| LP-DIXIT | L-8B | 0.000 | 0.565 | 0.435 | – | – | – |
| LP-DIXIT | M-7B | 0.000 | 0.522 | 0.478 | – | – | – |
| LP-DIXIT$_O$ | L-70B | 0.043 | 0.348 | 0.609 | – | – | – |
| LP-DIXIT$_O$ | L-8B | 0.000 | 0.304 | 0.696 | – | – | – |
| LP-DIXIT$_O$ | M-7B | 0.000 | 0.609 | 0.391 | – | – | – |
| LP-DIXIT$_D$ | L-70B | 0.130 | 0.565 | 0.304 | – | – | – |
| LP-DIXIT$_D$ | L-8B | 0.000 | 0.261 | 0.739 | – | – | – |
| LP-DIXIT$_D$ | M-7B | 0.043 | 0.739 | 0.217 | – | – | – |
| LP-DIXIT$_{OD}$ | L-70B | 0.217 | 0.435 | 0.348 | – | – | – |
| LP-DIXIT$_{OD}$ | L-8B | 0.130 | 0.391 | 0.478 | – | – | – |
| LP-DIXIT$_{OD}$ | M-7B | 0.087 | 0.609 | 0.304 | – | – | – |

**Table 2: Outcomes of LP-DIXIT on three KGs**

| KGE | LP-X | Mode | FB15k-237 | WN18RR | YAGO3-10 |
|---|---|---|---|---|---|
| TransE | DP | nec | 0.000 | 0.500 | **0.860** |
| | DP | suff | 0.010 | **0.510** | 0.860 |
| | Kelpie | nec | 0.000 | 0.020 | 0.280 |
| | Kelpie | suff | **0.030** | 0.000 | 0.580 |
| | GEnI | – | 0.010 | -0.050 | -0.020 |
| ComplEx | Criage | nec | **0.570** | 0.520 | 0.850 |
| | Criage | suff | 0.430 | 0.290 | 0.810 |
| | DP | nec | -0.020 | **0.600** | 0.700 |
| | DP | suff | -0.020 | **0.600** | 0.700 |
| | Kelpie | nec | 0.020 | 0.310 | 0.570 |
| | Kelpie | suff | 0.020 | 0.630 | 0.690 |
| | GEnI | – | -0.010 | -0.010 | -0.020 |
| ConvE | Criage | nec | 0.090 | **0.269** | 0.800 |
| | Criage | suff | **0.300** | 0.250 | **0.870** |
| | DP | nec | 0.000 | 0.038 | 0.070 |
| | DP | suff | -0.040 | 0.019 | 0.120 |
| | Kelpie | nec | 0.000 | 0.019 | -0.010 |
| | Kelpie | suff | 0.000 | 0.019 | 0.070 |

**Table 3: Outcomes of LP-DIXIT on KGs with schema**

| KGE | LP-X | Mode | Summ. | DB100K | DB50K | YAGO4-20 |
|---|---|---|---|---|---|---|
| TransE | DP | nec | – | 0.140 | 0.300 | **0.150** |
| | DP | suff | – | 0.110 | 0.300 | **0.150** |
| | Kelpie | nec | – | **0.150** | 0.330 | 0.080 |
| | Kelpie | suff | – | 0.080 | 0.310 | 0.140 |
| | GEnI | – | – | -0.040 | -0.020 | 0.030 |
| | Kelpie++ | nec | b | **0.150** | 0.300 | 0.080 |
| | Kelpie++ | nec | s | **0.150** | **0.340** | 0.070 |
| | Kelpie++ | suff | b | 0.130 | 0.300 | 0.080 |
| | Kelpie++ | suff | s | 0.140 | 0.320 | 0.120 |
| ComplEx | Criage | nec | – | **0.710** | **0.720** | 0.480 |
| | Criage | suff | – | 0.630 | 0.690 | **0.520** |
| | DP | nec | – | 0.460 | 0.300 | 0.110 |
| | DP | suff | – | 0.460 | 0.300 | 0.110 |
| | Kelpie | nec | – | 0.370 | 0.280 | 0.130 |
| | Kelpie | suff | – | 0.390 | 0.300 | 0.090 |
| | GEnI | – | – | 0.030 | -0.010 | 0.000 |
| | Kelpie++ | nec | b | 0.290 | 0.260 | 0.150 |
| | Kelpie++ | nec | s | 0.280 | 0.330 | 0.100 |
| | Kelpie++ | suff | b | 0.190 | 0.270 | 0.120 |
| | Kelpie++ | suff | s | 0.160 | 0.280 | 0.150 |
| ConvE | Criage | nec | – | 0.450 | **0.660** | 0.360 |
| | Criage | suff | – | **0.510** | 0.420 | **0.710** |
| | DP | nec | – | 0.270 | 0.090 | 0.060 |
| | DP | suff | – | 0.240 | 0.100 | 0.080 |
| | Kelpie | nec | – | 0.020 | 0.090 | 0.030 |
| | Kelpie | suff | – | 0.150 | 0.070 | 0.080 |
| | Kelpie++ | nec | b | 0.130 | 0.100 | 0.060 |
| | Kelpie++ | nec | s | 0.080 | 0.130 | 0.080 |
| | Kelpie++ | suff | b | 0.180 | 0.100 | 0.150 |
| | Kelpie++ | suff | s | 0.130 | 0.050 | 0.130 |

the larger LLM can help avoid the added complexity of computing the rank to construct $O$ and selecting the examples to build $D$.

We chose LP-DIXIT with Llama-3.1 for the comparative study of *post-hoc* LP-X methods because it performs best on FR200K, which is the most reliable benchmark since it is user-rated. Moreover,

Llama-3.1 proved to be much faster than Mixtral-8x. The comparative studies with the other setups are available in our GitHub repository. Hence, in Tab. 2 we report the results in terms of $\overline{y}$ of the comparative evaluation of the *post-hoc* LP-X methods on FB15k-237, WN18RR, and YAGO3-10, while in Tab. 3 we report the results of the evaluation, including Kelpie++, on DB50K, DB100K, and YAGO4-20. On all KGs, Criage and Data Poisoning performed best, even though these methods produce the simplest explanations: those consisting of a single triple. In contrast, GEnI leads to limited (often negative) FSV. We posit that this occurred because GEnI very often failed to generate an explanation, we handled such cases by running the post-explanation simulation in LP-DIXIT without an explanation, analogously to the pre-explanation simulation.

Moreover, in Tab. 4 we report the outcomes of LP-DIXIT$_O$ on DB50K, DB100K, and YAGO4-20. Note that the results between Tab. 3 and Tab. 4 are often very similar. This is because $O$ and/or $D$ are added to both the pre-explanation simulation and the post-explanation simulation, improving the simulation accuracy in both steps. To clarify, if both $\overline{s}$ and $\overline{s}^X$ increase by the same amount, the difference between them remains the same. Indeed, we report in Tab. 5 the values for $\overline{s}$ obtained by LP-DIXIT and LP-DIXIT$_O$ for each dataset and model; such a metric does not depend on the LP-X method and is thus essentially identical for all such methods: we report it only once for each model and dataset. Tab. 5 shows that LP-DIXIT$_O$ consistently improves on LP-DIXIT in performing pre-explanation simulations. Since $\overline{s}$ is higher for LP-DIXIT$_O$ than for LP-DIXIT, while $\overline{y}$ is close between them, it follows that $\overline{s}^X$ is also higher, i.e., LP-DIXIT$_O$ also improves on LP-DIXIT in performing

**Table 4: Outcomes of LP-DIXIT$_O$ on KGs with schema**

| KGE | LP-X | Mode | Summ. | DB100K | DB50K | YAGO4-20 |
|---|---|---|---|---|---|---|
| | DP | nec | – | 0.110 | **0.310** | 0.040 |
| | DP | suff | – | 0.120 | **0.310** | **0.050** |
| | Kelpie | nec | – | 0.120 | 0.260 | 0.000 |
| | Kelpie | suff | – | 0.030 | 0.270 | -0.020 |
| TransE | GEnI | – | – | 0.050 | -0.040 | -0.110 |
| | Kelpie++ | nec | b | 0.160 | 0.290 | 0.000 |
| | Kelpie++ | nec | s | 0.170 | 0.280 | -0.070 |
| | Kelpie++ | suff | b | **0.210** | 0.270 | -0.050 |
| | Kelpie++ | suff | s | 0.200 | 0.270 | -0.060 |
| | Criage | nec. | – | **0.570** | **0.580** | **0.380** |
| | Criage | suff | – | 0.420 | 0.520 | 0.360 |
| | DP | nec | – | 0.320 | 0.240 | -0.010 |
| | DP | suff | – | 0.320 | 0.240 | -0.010 |
| ComplEx | Kelpie | nec. | – | 0.180 | 0.200 | 0.070 |
| | Kelpie | suff | – | 0.150 | 0.240 | 0.040 |
| | GEnI | – | – | -0.020 | 0.000 | -0.010 |
| | Kelpie++ | nec | b | 0.190 | 0.170 | 0.030 |
| | Kelpie++ | nec | s | 0.190 | 0.250 | 0.050 |
| | Kelpie++ | suff | b | 0.130 | 0.240 | 0.100 |
| | Kelpie++ | suff | s | 0.090 | 0.200 | 0.100 |
| | Criage | nec | – | 0.350 | **0.480** | 0.210 |
| | Criage | suff | – | **0.430** | 0.300 | **0.330** |
| | DP | nec | – | 0.220 | 0.070 | -0.040 |
| | DP | suff | – | 0.190 | 0.080 | 0.070 |
| ConvE | Kelpie | nec. | – | -0.040 | 0.040 | 0.000 |
| | Kelpie | suff | – | 0.080 | 0.040 | 0.070 |
| | Kelpie++ | nec | b | 0.050 | 0.050 | 0.000 |
| | Kelpie++ | nec | s | 0.080 | 0.020 | 0.000 |
| | Kelpie++ | suff | b | 0.090 | 0.020 | 0.040 |
| | Kelpie++ | suff | s | 0.100 | -0.010 | 0.030 |

**Table 5: Performance in pre-explanation simulation**

| KGE | FSV | DB100K | DB50K | YAGO4-20 |
|---|---|---|---|---|
| TransE | LP-DIXIT | 0.203 | 0.080 | 0.130 |
| | LP-DIXIT$_O$ | **0.359** | **0.330** | **0.622** |
| ComplEx | LP-DIXIT | 0.130 | 0.080 | 0.130 |
| | LP-DIXIT$_O$ | **0.310** | **0.350** | **0.423** |
| ConvE | LP-DIXIT | 0.240 | 0.070 | 0.120 |
| | LP-DIXIT$_O$ | **0.360** | **0.400** | **0.530** |

post-explanation simulation. Thus, LP-DIXIT proved to focus on the evaluation of the explanations rather than on the LLM's ability to answer LP queries: if this ability increases but the explanations remain the same, the FSV does not change. The addition of $O$ and/or $D$ can be seen as making the verifier more resourceful and thus potentially more able to mimic human users in the FSV.

Moving to a qualitative analysis, we now illustrate typical examples of explanation output along with the results of the post-explanation simulation performed with Llama-3.1-70B. We report the explanations generated by DP (nec) and Kelpie (nec.) for predictions performed with TransE on YAGO3-10. We specifically focus on the triple ⟨*Ihor_Korotetskiy, isAffiliatedTo, FC_Shakhtar_Donetsk*⟩. The explanation generated by DP is the triple ⟨*Ihor_Korotetskiy, playsFor, FC_Shakhtar_Donetsk*⟩. In contrast, the explanations generated by Kelpie consists of 4 triples: ⟨*Ihor_Korotetskiy, playsFor, FC_Shakhtar_Donetsk*⟩, ⟨*Ihor_Korotetskiy, isAffiliatedTo, FC_Illichivets_Mariupol*⟩,

⟨*Ihor_Korotetskiy, isAffiliatedTo, FC_Kryvbas_Kryvyi_Rih*⟩, and ⟨*Ihor_Korotetskiy, playsFor, FC_Zorya_Luhansk*⟩

The post-explanation simulation in the case of DP is the ground truth *FC_Shakhtar_Donets*, whilst in the case of Kelpie it is *FC_Illichivets_Mariupol*. It seems that the additional triples in the explanation by Kelpie mislead the model whilst the explanation by DP being simpler potentially include less misleading information.

## 6 Conclusion

We introduced LP-DIXIT, a novel approach to algorithmically evaluate explanations considering the perspective of users. Specifically, it determines the FSV by employing LLMs to mimic human verifiers. We performed an experimental evaluation on two existing benchmarks to assess the alignment of LP-DIXIT with human judgment. LP-DIXIT was also developed to address the lack of comparative studies of *post-hoc* LP-X methods.

Whilst the results demonstrated the effectiveness of LP-DIXIT, some limitations of our approach deserve further investigation. First, in our formalization of the FSV a simulation is correct if the entity returned by the verifier is equal to the prediction of the KGE that is a ground truth in the context of FSV. However, if the simulation differs from the ground truth, it may still lead to a true triple existing in the KG. Therefore, we intend to investigate the impact of distinguishing between the different types of errors in the simulation. Moreover, if an LP-X method fails to generate an explanation, LP-DIXIT executes the post-explanation simulation with no explanation, analogously to the pre-explanation simulation. We plan to further investigate the impact of failures on the overall performance. In addition, LP-DIXIT$_O$ keeps the set of possible entities ordered as obtained from the KGE and the demonstrations in LP-DIXIT$_D$ are preserved in the order obtained from filtering the triples. It may be worthwhile to assess the influence of different orders.

A natural extension of this work would be a formalization of the FSV with a more fine-grained output, e.g, in a continuous interval. For example, we could consider the simulation as a ranking of entities as fillers and then measure the ranking correlation with the ground truth ranking given by the KGE model. In addition, we plan to investigate on the use of RAG models as verifiers. Another goal may be experimenting with more advanced prompt engineering techniques, such as Chain-of-Though prompting. Finally, we could also conduct a user study of the FSV with human users as verifiers to gain additional insight into the alignment of LP-DIXIT with human judgment.

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

## A  Appendix: Hyper-parameters

In this appendix, we report in Tab. 6 the hyper-parameters that we adopted to train each KGE model on each KG and benchmark. Furthermore, we employed the same set of hyper-parameters to execute CRIAGE, DP, KELPIE, KELPIE++ and GENI to generate explanations.

Note that:

- $D$ is the embedding dimension which is identical for entity and relation embeddings in the models that we adopted
- $p$ is the exponent of the $p$-norm
- $Lr$ is the learning rate
- $B$ is the batch size
- $Ep$ is the number of epochs
- $\gamma$ is the margin in the *Pairwise Ranking Loss*
- $N$ is the number of negative triples generated for each positive triple
- $\omega$ is the size of the convolutional kernels
- $Drop$ is the training dropout rate, specifically:
  - $in$ is the input dropout
  - $h$ is the dropout applied after a hidden layer
  - $feat$ is the feature dropout

We adopted *Random Search* to find the values of the hyper-parameters, except for $B$ and $Ep$; the performance of each configuration is assessed on the validation set. Specifically, for $B$ we adopted the value 16536 for all configurations as it leads to optimize execution times and parallelism, exceptions are ComplEx and ConvE on FRUNI where we adopted 4096 as such KGs have a much more higher of entities and thus require more memory. While, for $Ep$ we adopted early stopping with 1000 as maximum number of epochs, 5 as patience threshold, and evaluating the model on the validation set every 5 epoch during the training of the models. Then, we reported the epoch on which the training stopped.

In all the LLMs we adopted the value 0.6 for the temperature parameter. Finally, we specify the value 0.6 for the threshold parameter required in GENI as it is one of the values suggested in [2].

**Table 6: Hyper-parameters of the KGE models**

| | | FB15k-237 | WN18RR | YAGO3-10 | DB50K | DB100K | YAGO4-20 | FRUNI | FR200K |
|---|---|---|---|---|---|---|---|---|---|
| TRANSE | $D$ | 256 | 128 | 256 | 128 | 256 | 64 | 64 | 128 |
| | $p$ | 2 | 2 | 2 | 2 | 2 | 2 | 2 | 2 |
| | $Ep$ | 90 | 160 | 70 | 180 | 215 | 100 | 30 | 65 |
| | $Lr$ | 0.008 | 0.014 | 0.042 | 0.001 | 0.026 | 0.008 | 0.002 | 0.028 |
| | $\gamma$ | 1 | 10 | 2 | 2 | 10 | 2 | 1 | 10 |
| | $N$ | 10 | 5 | 15 | 10 | 5 | 15 | 5 | 5 |
| CONVE | $D$ | 200 | 200 | 200 | 200 | 200 | 200 | 200 | 200 |
| | $Drop.in$ | 0.1 | 0 | 0.1 | 0 | 0.2 | 0.2 | 0.1 | 0.2 |
| | $Drop.h$ | 0.1 | 0 | 0 | 0 | 0 | 0.1 | 0 | 0.5 |
| | $Drop.feat$ | 0 | 0.3 | 0 | 0 | 0.2 | 0.3 | 0.2 | 0.2 |
| | $Ep$ | 270 | 50 | 565 | 65 | 670 | 535 | 30 | 45 |
| | $Lr$ | 0.021 | 0.029 | 0.012 | 0.023 | 0.034 | 0.042 | 0.037 | 0.016 |
| COMPLEX | $D$ | 256 | 265 | 256 | 256 | 64 | 64 | 256 | 256 |
| | $Ep$ | 124 | 239 | 94 | 104 | 1000 | 754 | 164 | 89 |
| | $Lr$ | 0.044 | 0.046 | 0.034 | 0.050 | 0.008 | 0.004 | 0.004 | 0.048 |

