# OpenReview forum: "LP-DIXIT: Evaluating Explanations for Link Prediction on Knowledge Graphs using Large Language Models"
_ACM.org/TheWebConf/2025/Conference — WWW 2025 Poster_

### Official Review · Reviewer_TGNy · 2024-11-28

**Novelty:** 4
**Technical Quality:** 4

**Review:**

The paper introduces LP-DIXIT, an algorithmic method for evaluating explanations of link predictions in knowledge graphs. Unlike traditional methods, which rely on simplistic benchmarks or user studies that suffer from scalability issues, LP-DIXIT uses forward simulatability variation to measure how explanations improve predictability. The method leverages large language models (LLMs) to mimic user responses and assess the effectiveness of explanations in a scalable and reproducible manner. The paper demonstrates that LP-DIXIT provides effective evaluations and compares it with existing state-of-the-art explanation methods.

**Questions:**

Limited Experimental Details
While the experiments prove the effectiveness of LP-DIXIT, the paper could provide more detailed analysis of how it compares to other methods, especially in edge cases or highly complex scenarios. More insights into how LP-DIXIT handles various graph structures or types of inferences would provide a clearer picture of its strengths and weaknesses.

Explanation Quality Measurement
The paper focuses on predictability (simulatability) as a measure of explanation quality, but the holistic quality of the explanations (such as their relevance, clarity, or completeness) is not addressed in depth. A more multifaceted approach to measuring explanation quality could provide a more comprehensive evaluation.

Adoption of New Metrics
While the forward simulatability variation metric is interesting, its adoption as a standard for evaluating explanations is still novel. The paper would benefit from a broader discussion on how this metric compares to existing evaluation metrics in the literature and whether it adequately captures all aspects of effective explanations.

**Reviewer Confidence:**

3: The reviewer is confident but not certain that the evaluation is correct

**Scope:**

3: The work is somewhat relevant to the Web and to the track, and is of narrow interest to a sub-community

---

### Official Review · Reviewer_MNfY · 2024-12-01

**Novelty:** 2
**Technical Quality:** 3

**Review:**

This paper introduces LP-DIXIT, a framework for evaluating the quality of explanations in link prediction tasks on knowledge graphs by quantifying how explanations improve users' ability to predict model outcomes (FSV).  While the problem is significant, the methodology lacks novelty, as the combination of LLM prompting and post-hoc explanation techniques has been explored previously. Furthermore, potentially impactful ideas such as RAG integration are mentioned as future work rather than being implemented. Besides, the paper suffers from numerous typos and formatting issues, including missing punctuation (e.g., lines 579 and 370), inconsistent table alignment (Tables 1–3), and improperly formatted sections (lines 858–870).

**Questions:**

- In Section 5.1, *“We kept only high-scoring explanations because neutral scores can lead to different opinions among users; similarly, low-scoring explanations may be unclear and ambiguous, leading to subjective judgments, whereas high-scoring explanations are likely to be less ambiguous, leading to more consistent judgments.”* Is this approach based on prior work, or was it introduced specifically for this paper’s task? Please clarify the rationale.

- In line 397, what does "simulating the inference leading to o^1” mean? This phrase lacks context, making it difficult to interpret its role in the methodology.

- The paper notes that *“when using Llama-3.1-70B, which is the largest model in terms of parameter size, the addition of O and D is almost always detrimental.”* Given that larger LLMs typically exhibit stronger in-context learning abilities, how do you explain this contradictory finding? Are there any specific limitations of Llama-3.1-70B on this task or your prompting approach that could account for this?

**Reviewer Confidence:**

3: The reviewer is confident but not certain that the evaluation is correct

**Scope:**

4: The work is relevant to the Web and to the track, and is of broad interest to the community

---

### Official Review · Reviewer_ExB6 · 2024-12-01

**Novelty:** 6
**Technical Quality:** 4

**Review:**

The paper presents LP-DIXIT, an approach to evaluating link prediction explanations (post-hoc) in knowledge graphs using LLMs. In particular, LLMs are applied to determining "Forward Simulatability Variation" (FSV), which measures how the provision of an explanation (traditionally to a human user) impacts the predictability of (ML model) output given an input. The results indicate that LP-DIXIT compares reasonably well with human user-generated benchmarks.

The paper covers a very relevant challenge for the Web and related technologies, and relates to important trends such as explainability and generative AI. It is well motivated and reasonably well-written. The evaluation is conducted using open-source LLMs, thus facilitating reproducibility.  A set of limitations is clearly stated, e.g. regarding the handling of simulation errors (or error types). The source code for the implementation and experiments is shared and includes documentation as well as additional results that did not make it into the paper. While I did not try to run the experiments, the code and documentation look reasonable. Some issues are, here, that the code (incl. the main notebook) is not well-documented and in the notebook print-outs, we can see that errors occurred during the latest execution.


Some shortcomings are:

- It may make sense to highlight the use of instruction-tuned LLMs more prominently. More importantly, it is not clear how the LLMs are instruction-tuned. If the LLMs are instruction-tuned, the tuning process is a crucial part of the approach and evaluation.

- One key limitation is an obvious issue with the general idea of using LLMs in order to mimic users for automatic evaluations of algorithms and software systems. The results of such evaluations are, clearly, proxy metrics that must be interpreted with extreme caution. One would expect that performance can vary substantially, depending (e.g.) on use-case and domain. While the last sentence of the conclusion relates to this issue, it may be good to discuss it explicitly.

- It would be good to discuss the evaluation results in greater detail, in particular because the tables are not easily interpretable. What exactly do the "best" (bold) values mean? What are the conclusions we can draw from them?

Minor comments:
- Abstract. The sub-sentence "We experimentally prove that LP-DIXIT evaluates as effective explanations those in benchmarks," is difficult to parse.
- Introduction: it is a very broad (probably too broad) claim that the OWA is "made by default in Logic", as the field of Logic is, arguably, very broad.
- "that can accommodating explanations with diverse
structures" -> "that can accommodate explanations with diverse
structures"
- "We also compare different LLM of different parameter size" -> "We also compare several LLMs of different parameter sizes"?
- "into an explanation, valuable explanations should be easy" -> "into an explanation. Valuable explanations should be easy"? More broadly, some sentences could be shortened for clarity by starting new sentences after the comma.
- "Kepple++ [...] ground" -> "Kepple++ [...] grounds"
- Bullet list just before Subsec. 4.2: full stop, semicolons and/or similar punctuation marks are missing
- "In such Table" -> In this table"

**Questions:**

1. How exactly was instruction tuning carried out?
2. Why does the example notebook contain errors stemming from its last run?
3. It would be good to discuss the evaluation results in greater detail, in particular because the tables are not easily interpretable. What exactly do the "best" (bold) values mean? What are the conclusions we can draw from them?

**Reviewer Confidence:**

3: The reviewer is confident but not certain that the evaluation is correct

**Scope:**

4: The work is relevant to the Web and to the track, and is of broad interest to the community

---

### Official Review · Reviewer_uSkB · 2024-12-02

**Novelty:** 6
**Technical Quality:** 4

**Review:**

This work evaluates the quality of explanations generated by link prediction models, rather than focusing on a specific link prediction technique itself. It employs a Large Language Model (LLM)-based approach to explain link prediction (triples) in knowledge graphs. Specifically, it leverages Forward Simulatability Variation (FSV), a metric that measures the variation in users' ability (simulated by large language models) to correctly predict links after receiving an explanation. The idea is to assess whether an explanation improves, has no effect on, or even deteriorates the "user's" ability to predict whether a link should exist. The work incorporates LLMs as a tool to simulate how a human user might understand and utilize the explanations provided by the link prediction model. These LLMs act as verifiers attempting to simulate link inference with and without the aid of provided explanations.


# Strengths
- Formalization: Formally describes the link prediction problem in knowledge graphs.
- Reproducibility: Source code and datasets are openly available.
- Relevance: The transparency of results obtained by learning models is of special importance, and this work advances in the direction of adding greater explainability.
- Innovation: Considers the use of large language models (LLMs) to simulate human comprehension in the evaluation of explanations.

# Weaknesses
- Scope: The proposal requires defining a set of candidates for the LLM-based system to choose from. It is a Multiple Choice Question-Answering task, rather than entity generation.
- Generalization: The method's ability to generalize to different types of knowledge graphs or to configurations where a candidate set cannot be easily defined is limited, which would prevent the reuse of this work in more open and less controlled scenarios.
- Clarity: The article's writing assumes certain concepts that are not necessarily familiar to the reader, which hinders fluid reading of the article (e.g., lines 593-595).

**Questions:**

- In what aspects does a RAG-based approach add complexity compared to the proposal in this work?
- Given that the prompting-based strategy limits the context size that an LLM can handle, has the possibility of incorporating KG knowledge through fine-tuning been evaluated?
- Since biases have been demonstrated to exist in LLMs, how could these be measured in these evaluations?

**Reviewer Confidence:**

2: The reviewer is willing to defend the evaluation, but it is likely that the reviewer did not understand parts of the paper

**Scope:**

3: The work is somewhat relevant to the Web and to the track, and is of narrow interest to a sub-community